# Into the Deep (Sequence) of the Foot-and-Mouth Disease Virus Gene Pool: Bottlenecks and Adaptation during Infection in Naïve and Vaccinated Cattle

**DOI:** 10.3390/pathogens9030208

**Published:** 2020-03-12

**Authors:** Ian Fish, Carolina Stenfeldt, Rachel M. Palinski, Steven J. Pauszek, Jonathan Arzt

**Affiliations:** 1Foreign Animal Disease Research Unit, Plum Island Animal Disease Center, ARS, USDA, Orient, NY 11957, USA; ian.fish@usda.gov (I.F.); carolina.stenfeldt@usda.gov (C.S.); rpalinski@vet.k-state.edu (R.M.P.); steve.pauszek@usda.gov (S.J.P.); 2Oak Ridge Institute for Science and Education, PIADC Research Participation Program, Oak Ridge, TN 37830, USA; 3College of Veterinary Medicine, Kansas State University, Manhattan, KS 66506, USA

**Keywords:** Foot-and-mouth disease virus, fmdv, fmd, quasispecies, population dynamics, within-host evolution, bottleneck, pathogenesis, aphthovirus

## Abstract

Foot-and-mouth disease virus (FMDV) infects hosts as a population of closely related viruses referred to as a quasispecies. The behavior of this quasispecies has not been described in detail in natural host species. In this study, virus samples collected from vaccinated and non-vaccinated cattle up to 35 days post-experimental infection with FMDV A24-Cruzeiro were analyzed by deep-sequencing. Vaccination induced significant differences compared to viruses from non-vaccinated cattle in substitution rates, entropy, and evidence for adaptation. Genomic variation detected during early infection reflected the diversity inherited from the source virus (inoculum), whereas by 12 days post infection, dominant viruses were defined by newly acquired mutations. Mutations conferring recognized fitness gain occurred and were associated with selective sweeps. Persistent infections always included multiple FMDV subpopulations, suggesting distinct foci of infection within the nasopharyngeal mucosa. Subclinical infection in vaccinated cattle included very early bottlenecks associated with reduced diversity within virus populations. Viruses from both animal cohorts contained putative antigenic escape mutations. However, these mutations occurred during later stages of infection, at which time transmission is less likely to occur. This study improves upon previously published work by analyzing deep sequences of samples, allowing for detailed characterization of FMDV populations over time within multiple hosts.

## 1. Introduction

Foot-and-mouth disease (FMD) is a viral disease affecting even-toed ungulates, causing economically devastating effects on animal production and international trade [1,2,3]. Though Europe and the Americas have largely eliminated FMD, the disease remains a substantial concern for livestock farmers in much of the world [4]. The clinical form of FMD manifests with fever, lameness, and characteristic vesicular lesions on the feet, oral cavities, and teats, thereby negatively impacting animal welfare and herd productivity [2,5]. The economic impacts associated with decreased production, disease surveillance, trade restrictions, and vaccination campaigns comprise a large burden to FMD-endemic and neighboring countries [3].

The etiologic agent of FMD, foot-and-mouth disease virus (FMDV) (family: *Picornaviridae,* genus: *Aphthovirus*) is a single-stranded positive-sense RNA virus with a particularly high mutation rate [6]. The approximately 8.3 kb genome includes a 7 kb open reading frame encoding a polyprotein that is post-translationally processed into four structural and eleven non-structural proteins including two forms of Leader protease (Lpro) and three copies of VPg [5,7]. High levels of variation, often associated with positive selection and antigenic escape, are commonly detected in the capsid protein coding regions [8,9,10]. The GH loop in capsid protein VP1 is particularly important for host cell entry and antibody-mediated neutralization [11]. Specifically, the Arg-Gly-Asp (RGD) motif within this GH loop interacts directly with the host cell integrin receptors (e.g., αvβ6) (reviewed in [12]).

Like many RNA viruses, FMDV naturally exists as a population with a complex depth of genetic variation, i.e., as a quasispecies. In recent years, next-generation sequencing technology has enabled subconsensus-level characterization of this complex viral population [13,14,15,16,17]. Viral population diversity and availability of low frequency mutations have been demonstrated to mediate viral swarm adaptability, fitness, and ultimately virulence [18,19]. For FMDV specifically, these quasispecies characteristics have been extensively investigated in cell culture (reviewed in [20]). Relating understanding of the FMDV quasispecies in vitro to swarm behavior in natural host species is an important goal as it relates to infection progression, differential virulence, and mechanisms mediating immunity in vaccinated and non-vaccinated hosts.

The path to improved FMD countermeasures may depend upon elucidation of viral and host determinants of FMDV evolution. Delineating differences in viral genetic change in response to both primed (vaccinated) and unprimed host humoral immunity are of particular interest as conventional FMD vaccines do not prevent subclinical or persistent infection. In cattle, both primary and persistent infection have been localized to the nasopharyngeal mucosa [21,22,23,24]. In non-vaccinated animals, acute FMDV infection lasts approximately one to two weeks, and involves systemic dissemination of virus and transient viremia [25]. The clinical phase of the disease is followed by a transitional period during which the virus is completely cleared in a subset of animals (herein referred to as terminators) [24]. Persistent FMDV infection is defined by the presence of infectious FMDV in oropharyngeal fluid (OPF) samples four or more weeks after infection [26]. Appropriately vaccinated animals are generally protected against systemic generalization of virus and clinical FMD. However, these vaccinated animals often undergo neoteric subclinical infection [27] and traverse the corresponding phases of early, transitional, and persistent infection, during which viral replication is fully restricted to the nasopharynx (upper respiratory tract). 

The present study builds upon previous work [16] by investigating the evolution of FMDV populations within natural hosts by examining the deep sequences of viruses sampled through 35 days following experimental infection of naïve and vaccinated cattle [23,24]. The results of these analyses suggest that vaccination causes narrower bottlenecks of FMDV populations compared to the viruses present in their non-vaccinated counterparts and that antigenic escape along with other novel mutations, occur during the persistent phase of infection. Previous findings of the transmission, maintenance and emergence over time of multiple haplogroups inherited from the inoculum were confirmed using a greater quantity of samples. These findings enhance understanding of FMDV evolution in vivo and may contribute to development of improved FMD vaccines. 

## 2. Results

### 2.1. Animal Experiments and Clinical Outcomes 

The 10 non-vaccinated cattle included in the study all developed fulminant clinical FMD after virus exposure. The 10 vaccinated animals were protected from clinical disease but were subclinically infected, as demonstrated by the repeated recovery of the virus from clinical samples. Details of clinical symptoms, infection dynamics, and tissue distribution of FMDV in these cattle have been published previously [23,24]. Half of the animals (five vaccinated and five non-vaccinated) were euthanized between 1 and 10 days post inoculation (dpi) for harvest of tissue samples (Appendix A) while the remainder of the animals were sampled through 35 dpi. Among the 10 cattle followed through study-end, seven were determined to be persistently infected with FMDV (carriers), while three individuals (animal IDs 14-108, 14-111, and 14-57) fully cleared infection (terminators) during the transitional phase of infection.

### 2.2. Effects of Vaccination on FMDV Populations

The rate of FMDV genomic change over time was compared between vaccinated and non-vaccinated cohorts in order to investigate if vaccination induced selective pressures upon the inoculating virus population that were distinct from those that occurred in naïve (non-vaccinated) animals. In order to quantitate this effect, pairwise differences over time were first calculated between each FMDV sample’s consensus sequence and the preceding sequence obtained from the same animal (inoculum used as 0 dpi). Though this is a consensus-level measure of genomic change, and thus does not account for the dynamics of FMDV populations (shifting subconsensus single nucleotide variants (SNVs)), this analysis is still valuable in order to identify changes in dominant genomes, utilize the full breadth of samples in this dataset, and relate this study to previous publications.

Viruses in non-vaccinated animals had higher substitution rates than the viruses in the vaccinated cohort during early (0.188 vs. 0.131 substitutions/site/year (subs/st/yr)) and transitional (0.127 vs. 0.089 subs/st/yr) phases of infection; however, these differences between groups were not statistically significant (Table 1). While the substitution rates decreased with advancing phase of infection for synonymous and nonsynonymous sites (Table 1, Appendix A), this change was only statistically significant for the non-vaccinated cohort between early infection (0.188 subs/st/yr) and persistent infection (0.080 subs/st/yr, *P* < 0.05). Specifically, nonsynonymous changes had the highest observed substitution rate change between early and persistent phases of infection (Appendix A, *P* < 0.005). These data suggest that the non-vaccinated cattle maintain a large, diverse, and dynamic virus population during early infection which evolves more slowly during the later disease stages. During the persistent phase of infection, FMDV genomic rates of change in non-vaccinated and vaccinated cattle were similar.

### 2.3. Entropy

Deep sequence analyses of the sampled FMDV populations allowed for estimation of Shannon entropy. This provided a site-specific quantitation of nucleotide variation for each sampled virus, calculated from aligned sequencing reads and averaged across distinct genomic regions. Entropy for the protein coding region (CDS) of the inoculum was 0.0156, which was greater than that of nearly all vaccinated cattle samples, indicating higher diversity (mean 0.0120, median 0.0110, Table 1 and Figure 1). This evidence aligns with a previous analysis of the multi-haplotypic composition of this inoculum [16]. The CDS average entropy across all samples from non-vaccinated cattle (mean 0.0171, median 0.0169), was similar to the inoculum. The average dpi-matched non-vaccinated sample entropies were significantly (*P* < 0.001) higher than samples from vaccinated cattle across the full CDS, as well as for the capsid and nonstructural coding regions separately (Table 1). Since coding region entropies did not significantly change within either cohort over time (Figure 1), non-vaccinated cattle maintained significantly higher average entropies than vaccinated cattle through all phases of infection (Table 1). These data suggest a strong, early, and enduring reduction in FMDV population diversity, i.e., a narrow bottleneck, in the vaccinated animals. In contrast, there was little or no evidence of a reduction in diversity or effective population size during initial infection of non-vaccinated cattle (Table 1 and Appendix A).

Average site-wise entropy across the capsid coding regions for VP1, VP2, and VP3 was calculated in proportion to the overall CDS entropy (capsid / CDS entropy, Table 1); this allowed for comparison between samples of capsid entropy proportional to each sample’s global (CDS) diversity. Averaged across all phases of infection, capsid/CDS entropy values of the viral populations were significantly higher in non-vaccinated animals (1.17, *P* < 0.001) than in vaccinated animals (1.08, Table 1). This suggests that the early immune response in the vaccinated animals more strongly reduced relative capsid diversity in these hosts and that this took place rapidly following infection. This was also consistent with population bottlenecks, though it does not directly imply any specificity with regards to immune response (e.g., antibody-mediated neutralization). Notably, there was no significant difference in entropy or relative capsid entropy based upon the phase of infection (time) within either cohort (*P* > 0.05). This suggested that reduced FMDV diversity in vaccinated animals was sustained through the phases of infection examined herein. For non-vaccinated animals, this suggests that substantial population diversity is maintained despite the reduction in total virus load associated with the clearance of generalized infection and virus restriction to the nasopharynx during persistent infection. This is consistent with previous reports of FMDV population diversity detected during persistent infection [28,29].

### 2.4. Haplotypic Population Structure

The inoculum used to infect the animals in this study was derived from pooled vesicular lesion samples from multiple cattle which resulted in a highly heterogeneous virus population. As previously reported for a subset of the current samples, multiple haplogroups originating in the inoculum were detected at consensus level within samples derived from different animals at different times after infection [16]. The phylogenetic relationship between these viruses was assessed by maximum likelihood (Figure 2) and six haplogroups (A through F) were assigned based on phylogenetic clustering and inferred ancestral relationships.

Polymorphisms present at ≥2% (Appendix A) were assessed for all deep-sequenced samples (75 of 103 total samples; see Appendix A). Sets of single nucleotide polymorphisms (SNPs) characteristic of haplogroups A–F were used to classify subpopulations present within samples (Figure 3). Non-vaccinated cattle’s acute phase virus populations were highly haplotypically diverse, while vaccinated host samples tended to include only single haplotypes. In addition to haplotypic polymorphism, abundant variation ≥2% was regularly detected in virus populations (Appendix A). Overall, substantially less genetic diversity was detected in samples derived from vaccinated animals as compared to non-vaccinated animals. Samples isolated from non-vaccinated hosts were in daily flux through the early days of infection and regularly included viruses belonging to multiple haplogroups. For example, in animal 14-34, group A viruses dominated samples from 1 and 2 dpi, group B and F viruses co-dominated at 3 dpi and group F viruses dominated at 4 dpi (Figure 3). This is consistent with the high entropy and elevated rates of substitution measured for these animals (Table 1, Figure 1 and Appendix A). In contrast, virus populations in vaccinated cattle typically contained a single haplogroup and less polymorphism overall (Figure 3 and Appendix A).

Considering the available data, as both animal cohorts (vaccinated and non-vaccinated) progressed to the persistent phase of infection, within-host haplotypic diversity decreased (Figure 3). In nearly all cases, viruses detected during the carrier phase appear to belong to singular haplotypes. The one exception to this was animal 15-13’s persistent phase samples that sequentially included viruses belonging to either groups B or F. Across animals, no specific haplogroup dominated for any particular time range or cohort, thus no objective fitness advantage was inferred between haplogroups. Despite haplotypic stabilization in later stages of infection, population diversity as indicated by subconsensus polymorphism did not decrease over time (Appendix A). Rather, population diversity was either maintained or increased through the persistent phase of infection. Specifically, OPF samples at 28 dpi from animals 14-34, 14-110, 15-12, and 15-14 all included substantial assortments of SNPs at frequencies between 10% and 49% (11–24 SNPs each, Appendix A). For persistent phase samples denoted as ‘undetermined’, this category has been assigned in cases where only consensus-level data was available (in which case only >50% of variation was assignable to a specific haplogroup), or when one of the six determined haplogroups could not be assigned to the subconsensus variation. For the latter, this may result from an undefined clade, reversion, recombination, or other undetermined processes.

### 2.5. Genomic evolution of the Viral Swarm

The site-specific heterogeneity within the initial infecting population (inoculum) was assessed and compared to consensus sequences of samples collected from the infected animals. Across the inoculum CDS, ultra-deep sequencing (10.4 million reads) indicated 217 variable sites encoding 220 single nucleotide variants (SNVs) at frequencies ≥0.5%, three of which had multiple polymorphisms at the same site (Appendix A). Twenty-seven (12.3%) of these SNVs were identified at the consensus level in cattle samples (Figure 4). All SNPs that were present at high frequency (>10%) in the inoculum, 14 in total, were detected at the consensus level in multiple samples (Appendix A); this is consistent with genetic drift contributing to the dominance of specific genotypes. As a gross means of measuring the influence of ancestral variation compared to novel mutation over time, inoculum minority variants (≥0.5%) present in sample consensus sequences (ancestral SNPs) on each sampling date were measured proportionally to those SNPs not detected in the inoculum (novel SNPs) (Figure 5). Within the first week of infection, the majority of SNPs in virus consensus sequences were ancestral, i.e., already present in the founding populations. The rate of loss of ancestral SNPs and gain of novel SNPs were comparable between non-vaccinated and vaccinated cattle (not shown). At approximately 10 days after infection, novel SNPs became equally common to ancestral SNPs. 

### 2.6. Nonsynonymous Substitutions 

Genetic variations that resulted in amino acid substitutions in sampled viruses were examined at the consensus and subconsensus level. Focusing on the regions encoding the capsid proteins VP2, VP3, and VP1 at the consensus level, 31 sites encoded amino acid changes yet only three of these were polymorphic in the inoculum (Figure 4 and Appendix A). Homology modeling implicated 15 of these sites as putative antigenic targets based on published data [31,32,33,34,35,36,37]: VP2-78, 82, 88, and 131; VP3-70, 131, and 175; VP1-96, 142, 144, 147, 155, 196, 197, and 199 (Figure 6). Many of these have been identified and discussed previously [16]. In order to detect residues with evidence of positive selective pressure, mixed effects model of evolution (MEME) analysis was run collectively (per host) on FMDV haplotypes reconstructed from serially-sampled deep sequence [38]. Haplotypes present in each sample were resolved with ViQuaS quasispecies reconstruction pipeline [39]. Because ViQuaS can resolve haplotypes within a sample population at exceedingly small frequencies (<<0.005) if provided rich read depth, the frequencies of haplotypes utilized for downstream analysis were cut-off at >0.5% or >2.0%. The presence of positively-selected amino acid changes predicted my MEME (*P* ≤ 0.10) are tallied for each host (Figure 4). The majority of consensus-level nonsynonymous substitutions and those detected in MEME analysis were detected only transiently (did not become fixed) and none were detected consistently in either vaccinated or naïve groups. 

The greatest quantity of residues with evidence of positive selective pressure was in VP1, specifically in the GH loop (Figure 4, Figure 6). The canonical FMDV receptor in cattle is integrin αvβ6, which binds to a conserved RGD motif within the GH loop of VP1 [40,41]. The vaccine used in this study was an adenovirus-vectored recombinant which encoded RGD at the anti-receptor motif. However, the inoculum encoded a ser-gly-asp (SGD) motif at this locus (residues 144–146) with no evidence of RGD-encoding viruses ≥0.5% in the population (Appendix A). The earliest detection of RGD at this site was in animal 14-34 at 6 dpi, in which 4.4% of the virus population had the arginine substitution. The following day, 99.3% of the virus population sampled in this host encoded RGD at the GH loop. Ultimately, VP1-S144R came to fixation in all cattle sampled in the persistent phase except for two vaccinated individuals (15-12 and 15-13); this substitution was also commonly observed in persistent-phase cattle samples in a study that used the same inoculum [42]. The latest initial detection of a dominant RGD genome was in animal 14-33, in which it emerged between 10 dpi (≤2%) and 21 dpi (99.6%). The VP1-S144R substitution was separately encoded by two of the three possible S>R codon changes (AGU>CGU and AGU>AGA) in five of the six characterized haplotypic backgrounds and one uncategorized genotype (15-14 at 17 and 28 dpi). There was also evidence of multiple independent RGD-encoding subpopulations co-infecting hosts. The subconsensus variants in animal 14-110 samples at 14 and 17 dpi included dozens of intermediate-frequency (10%–50%) SNPs indicative of many different viruses alongside a fixed (99.8%) VP1-S144R substitution (Appendix A). At the consensus level, a distinct shift in dominance from one haplogroup (group B) to another (group F) was evident between 17 dpi and 21 dpi, differing by 23 SNPs (Figure 2). 

There was strong evidence suggesting that the selection for RGD viruses resulted in global reductions in population diversity and the introduction of novelty through genetic draft. The clearest evidence of this was the dominance of haplogroup C viruses in animals 14-34 and 14-49, in which a series of changes in the coding region for 2C - T43M…H84N…D136E…I248T co-emerged with VP1-S144R at corresponding proportions (Appendix A and Appendix A). Selective sweeps associated with RGD genome emergences were also evident in animal 15-14 at 17 dpi (VP3-A75V and VP1-I35V) as well as animal 14-33 at 21 dpi (VP1-G33S) (Appendix A and Appendix A). These sweeps may, in part, result from capsid-directed selective pressure in addition to receptor-associated fitness gain. In contrast, animals 15-12 and 15-13 maintained SGD viruses through study end. Interestingly, this was associated with consensus-level changes indicative of antigenic escape. In 15-12, 21 dpi FMDV samples had a qualitatively divergent VP2-E82K capsid substitution and subsequent 28 and 35 dpi samples had VP2-H88N and VP1-V155A substitutions. In 15-13, 21 and 28 dpi viruses had VP3-E131K capsid substitution and were followed at 35 dpi by variants with dominant VP2-E131G and VP3-E131G substitutions. Each of these amino acid changes involved electrostatic shifts on the capsid surface in known antigenic regions (Figure 6). 

In the present study, there were relatively few sites with evidence of adaptation within nonstructural proteins. The most common replacement in nonstructural regions found across all animals was 3A-N136D, which was present in 15 of 20 animals. Although this replacement was relatively common in the inoculum at 35.9%, its presence as fixed or in the final sample consensus of 5/7 persistently infected animals, suggested an adaptive advantage. MEME analysis identified several sites under selective pressure in coding region for the C-terminus of 3A, with 3A-N136D identified in more cattle than any other substitution (Figure 4). In contrast, other commonly variable consensus-level amino acid changes, such as 2C-V283M and 3A-S117N, had no evidence of adaptive value in that there was little or no predilection for fixation.

### 2.7. Persistent Infection & the Nasopharynx

The hypothesis that viral population diversity correlated with the establishment or maintenance of persistent FMDV infection was tested. Viruses characterized from terminators (14-108, 14-111, and 14-57) did not significantly differ from dpi-matched viruses sampled from persistently infected carriers in substitution rate (4.29 vs. 3.47 subs/day, Appendix A). CDS entropy, though lower in terminators near their final dates of virus detection, was not significantly below that of dpi-matched carriers (Figure 1). This difference was only statistically significant within the nonstructural protein coding regions, suggesting more conservation in these regions of the FMDV genomes in terminators than in carriers. While very few consensus-level amino acid changes were identified in non-carrier viruses, the mutation VP3-Q220R, located at the VP3-VP1 cleavage site and on the capsid surface, was uniquely identified in samples derived from these cattle at 10 dpi.

## 3. Discussion

Although the quasispecies character of RNA viruses has been known for many years, investigation during infection of natural hosts, considering the inter-relationships with hosts’ biological processes, has received less attention. In this study, samples collected from FMDV-vaccinated and non-vaccinated cattle over five weeks were analyzed by deep-sequencing in order to investigate viral population dynamics across the phases of infection. Host vaccination status was associated with significant differences in virus substitution rates, entropy, and evidence for adaptation. While both cohorts established and maintained FMDV infection at similar prevalence [24,43], evaluation of deep sequence clearly demonstrated narrow population bottlenecks during early infection of vaccinated animals in contrast with the absence of population contraction in non-vaccinated hosts. Specifically, measures of global Shannon entropy and rates of consensus-level nucleotide substitution were consistently lower through the early phase of infection in virus populations sampled from vaccinated versus non-vaccinated cattle (Table 1, Appendix A, Appendix A). The primary conclusions from previous work [16], that FMDV evolved within cattle via haplotypes changing dominance alongside continuous point-mutation, were corroborated. This analysis has refined the earlier conclusions to include co-dominance of haplotypes in acutely infected naïve cattle and evidence of population bottlenecks associated with vaccination and specific mutations in the receptor binding domain. Furthermore, the previous finding of clade (haplogroup) stabilization during persistent phase has been refined with the identification of intra-haplogroup subpopulations.

Additionally, virus populations in samples from early infection in non-vaccinated animals were composed of multiple haplogroups, while those in samples from vaccinated cattle belonged to single lineages. However, the endurance of the multiple founding lineages became increasingly uncommon in non-vaccinated hosts over time, and by 21 dpi, samples included only single haplogroups. This pattern is consistent with bottlenecks observed in the transition from acute to chronic stages of infection with hepatitis C virus [44] and human immunodeficiency virus 1 [45]. Those studies reported that acute infection ended with a bottleneck of the multiple lineages that had thus far co-existed, leaving only viruses of a single lineage during chronic infection. In the present study, this decrease in haplogroup heterogeneity during persistent infection was accompanied by intra-haplotypic diversification, as evidenced by abundant subconsensus polymorphism and sustained entropy measures.

The early phase of infection in both non-vaccinated and vaccinated hosts are situations in which virus adaptation is hypothesized to be low (reviewed in [46]). In naïve cattle, viruses are entering a new environment in which minimal immune response [23,43] allows for relatively unconstrained growth of all competent viruses, i.e., a context in which fitness differences between variants are minimized. In the current study, these conditions contributed to relaxed selective pressure which resulted in maintenance of inherited variation. In vaccinated cattle, a strong primed immune response [23] provided efficient restriction of most introduced viruses when compared to non-vaccinates, as illustrated by low entropy and low global variation. Importantly, this restriction eliminated low-frequency mutants with potential adaptive value. Thus, there was no evidence for viral adaptation in either cohort during the early stages of infection. This is consistent with a previous analysis of FMDV minority variance within serial samples taken from FMDV-infected cattle during the acute phase of infection which suggested genetic drift as the primary mechanism of FMDV evolution both within and between hosts [47]. Of importance to FMDV epidemiology, these findings suggest that variation detected in field samples from FMDV outbreaks (acute phase) is likely the result of stochastic processes, e.g., genetic drift and transmission bottlenecks. However, the present study utilized a highly diverse inoculum which was directly deposited at the site of primary infection. It is possible that these factors could have facilitated primary infection with a genomic diversity beyond that experienced through natural transmission events.

Nonetheless, virus adaptation did take place within the scope of this study, most notably in genomic regions encoding capsid proteins. The canonical FMDV receptor in cattle is integrin αvβ6, which the virus utilizes for host cell entry via a conserved RGD motif within the GH loop of capsid protein VP1 (residues 144-146) [40,41]. The inoculum used in the current set of experiments was known to instead encode an SGD motif at this locus [16] as a result of prior passages in bovine tongue epithelium (see methods for details). Mutation of the SGD motif to RGD is fitness-enhancing as it facilitates integrin binding and host cell entry. Specifically, experimental work has demonstrated that VP1-144 serine to arginine substitution allows for improved cell to cell transmission [48]. In the current study, RGD fixation took place in every non-vaccinated animal sampled after 10 dpi (5/5), yet in only half of vaccinated animals through the persistent-phase to the study end (2/4). This suggests that, even though vaccination does not prevent subclinical or persistent infection, it can provide improved protection from critical viral adaptations (in this study, SGD->RGD). This represents an important benefit not typically attributed to vaccines, i.e., impeding the accrual of mutations which might be beneficial to the virus. 

Experimental works have shown that escape mutations were likely to arise at GH loop residues near to the conserved RGD motif, including under GH loop-specific monoclonal antibody neutralization [49,50,51,52,53]. In the current study, MEME analysis identified numerous sites in antigenic capsid regions suggestive of escape mutations. The majority of the sites identified by MEME were detected at low subconsensus frequencies and were not shared across animals. This type of low-frequency variation, independent of vaccination status or phase of infection in this study, is consistent with mutant swarm character described as central to the concept of FMDV quasispecies [6,54]. 

In contrast to the SGD-RGD transformation which occurred in most animals, viruses isolated from vaccinated animals 15-12 and 15-13 never acquired the RGD motif nor were these populations affected by associated selective sweeps. Viruses in these two hosts were thus more capable of acquiring predicted escape substitutions featuring substantial electrostatic changes within the time frame of the study. Specifically, persistent phase viruses included replacements at VP2-82 and -88, VP3-131 and VP1-155. An alternative hypothesis for capsid mutation particular to SGD viruses is adaptation to an alternative host cell receptor, such as heparan sulfate [55]. However, evaluation of amino acid changes in capsid structural models did not support any of these lying in the heparan sulfate binding site (Figure 6 inset) [32]; nonetheless, efficiency to bind other integrins or alternate receptors may be involved [48,56]. The extent to which the recombinant RGD vaccine prevented emergence of the RGD motif in two vaccinated animals or imposed differential selective pressure on different variants could not be verified within the current study design. 

In six serially sampled cattle, MEME identified residue 136 in the 3A protein to have evolved under positive selective pressure. 3A protein is a membrane-integrated protein that interacts with FMDV RNA polymerase and, although not fully understood, has been implicated in intracellular transport [57,58]. Specific deletions in the C-terminus of 3A have been associated with clinical attenuation in cattle [59,60,61] while virulence in pigs is maintained [62,63]. 

In order to test the hypothesis that FMDV clearance was associated with specific virus population characteristics, samples from terminators were examined for consistent trends. Viruses isolated from these hosts during their later stages of infection had reduced global entropy when compared to both earlier samples from those same animals and comparable carrier viruses. Specifically, virus sampled from animal 14-108 at 10 dpi encoded a fixed haplotype (group F) with low entropy and no polymorphism; these population characteristics are consistent with a swarm that was nearing extinction. Viruses from animal 14-111 also had reduced global entropy without ever acquiring the VP1-144R that was detected in all other viruses isolated from non-vaccinated cattle, possibly making the virus more vulnerable to clearance despite its two divergent lineages (groups B and F). A single amino acid replacement shared exclusively by terminators was VP3 Q220R, which has previously been shown to be prone to variation, including glutamine and arginine [31,64], and thus unlikely deleterious, merits further investigation. These results suggest that reduced diversity of viral populations may contribute to termination of infection, consistent with the concept that the mutant spectrum is important to viral fitness [20,65]. Specific host immunological profiles [43] were not associated with observed viral genomic changes. The mechanisms responsible for FMDV persistence could not be established, but future investigation focused on the FMDV quasispecies and host factors during the transitional period in terminators may elucidate the critical factors. 

The particularly rapid mutation rates of RNA viruses can facilitate responses to changing adaptive host immunity through the selection for escape mutants at antigenic epitopes [66,67,68,69]. We hypothesized that as FMDV evolved within each host, the virus would acquire escape mutations as a result of actuated humoral and cellular immune responses. However, based upon comparative literature- and homology model-based SNP analysis, there was limited expansion of antigenic diversity observed in the majority (5/7) of carriers. Each of these five cattle acquired populations with fixed (>98%) RGD genomes associated with selective sweeps. Sweeps included the clearance of ancestral and novel low-frequency variation as well as genetic draft. Loss of low-frequency variation provides a plausible explanation for the limited detection of antigenic variation in these cattle while genetic draft may explain the acquisition of irregular changes (e.g., 2C substitutions in haplogroup C members) that occurred synchronously with RGD replacement. Although previous studies have demonstrated that continual changes to FMDV capsids occur over longer time courses of persistent infection [70,71], this could not be addressed in the current study.

Notably, there is some suggestion from the current findings that selective pressure on the virus may also be reduced during FMDV persistence. Specifically, capsid/CDS entropy did not significantly differ between phases of infection (in either cohort), nor did subconsensus or consensus amino acid replacements indicate strong adaptation. This is consistent with previous findings in Cape buffalo that autologous antibody neutralization of FMDV does not change throughout persistent infection [29] and other reports demonstrating that the nasopharyngeal mucosa may function as an immunoprivileged or immunosuppressed site, supported by gene expression patterns suggesting a down-regulated anti-viral response [72,73]. This privileged state may in effect relax selective pressure, thus further limiting persistent virus escape adaptation. An ongoing research goal is to integrate sub-anatomic host tissue features and signaling patterns with FMDV subconsensus variation.

Persistent FMDV infection in cattle is restricted to distinct epithelial foci within the nasopharyngeal mucosa [22,24,74]. If these foci represent distinct viral subpopulations, this would be expected to be reflected in the deep sequence of oropharyngeal fluid samples which are retrieved with a probang cup, which harvests cells from multiple regions of the pharynx epithelium. Each OPF sample included at least two subpopulations in all hosts. These subpopulations may represent sub-anatomic vicariance at distinct epithelial foci, i.e., viruses replicating in isolated groups, leading to genomically identifiable subgroups. Coexistent persistent-phase FMDVs belonging to different lineages has previously been reported for cattle [75,76]. Genomic RNA, and in some cases, infectious virus, belonging to multiple FMDV serotypes have been detected in subclinically infected Cape buffalo (*Syncerus caffer*) and water buffalo (*Bubalus bubalis*) [27,28,77]. Further, coexistent viruses are a prerequisite for recombination, which has been demonstrated to play a role in the evolution of FMDV [28,78,79]. In the present study, the divergence among sampled viruses was inadequate for recombination detection. Nonetheless, inability to determine haplogroups in late-stage samples from hosts 15-14 and 14-33 was due to ambiguity of clade-informative SNPs, which may have been a result of recombination. 

These findings have important implications for the inter-relationship between FMDV within-host evolution and transmission. In both vaccinated and non-vaccinated hosts, variation detected during the first few days of infection appears not to be driven by selective (immunological) pressures. Novel mutations, while highly-adaptive, took at least one week to reach consensus level. Because most transmission of FMDV is believed to occur within the first few days of infection [80,81], these novel mutations would have a narrow chance of being passed on within this window. If this course of evolution is typical throughout chains of transmission, it follows that nearly all FMDV genomic change observed in field isolates is the result of purifying and neutral evolution, as has been suggested for the virus [47,76,82,83]. For those low-frequency adaptive SNPs that are successfully transmitted, neutral or weak-purifying selection within this window would not favor them through extended chains of transmission. However, several aspects of FMDV evolution and transmission could not be addressed by the currently study design, e.g., the gap between the viral genomes sampled and sequenced and those responsible for encoding viral proteins or successfully transmitting to another host. These may be the subjects of subsequent investigation.

### Summary and Conclusion

Subconsensus variation in FMDV populations were investigated in vaccinated and naïve cattle for 35 days following simulated natural infections. FMDV genomic change detected during early infection was consistent with neutral evolution in all cattle. A critical capsid adaptation at the site of host cell entry, VP1 S144R, came to fixation in most animals and was associated with selective sweeps; putative antigenic escape mutations only arose in vaccinated animals within the time frame of the study. Furthermore, during early infection, vaccination caused virus population bottlenecks which did not occur in the naïve cattle. This differential quasispecies behavior in vaccinated hosts may provide insights into further enhancement of countermeasures to impede viral propagation at the individual animal and population levels. For instance, the elusive goal of creating vaccines that can provide sterile immunity and prevent primary infection might derive from engineering vaccines which are specifically designed to protect against a broader range of genomic variants. Additionally, multiple subpopulations were present in viruses recovered during the persistent phase, consistent with distinct foci of FMDV infection in nasopharyngeal epithelial cells, furthering our understanding of the nature of persistent infection. These findings contribute novel insights to the evolution of FMDV in natural host species. 

## 4. Materials and Methods 

### 4.1. Animal Studies 

The animal experiments were part of a multi-study analysis of the FMDV carrier state described in previous publications [23,24,43]. All studies were carried out at Plum Island Animal Disease Center, New York under BSL-3Ag conditions and with approval from the Plum Island Animal Disease Center Institutional Animal Care and Use Committee (protocol 209-13). Briefly, a group of steers were vaccinated with a recombinant adenovirus-vectored FMDV A vaccine 2-weeks prior to intra-nasopharygeal inoculation with FMDV A24 Cruzeiro. In parallel, a group of non-vaccinated animals were inoculated with the same FMDV A24 Cruzeiro inoculum. Animals were sacrificed at predetermined time points, up to 35 days post inoculation and tissues were harvested for analysis.

Three distinct phases, namely, the early, transitional, and persistent phases, define FMDV infection in livestock [24,84]. The phases vary between animals of different immune statuses, i.e., vaccinated or non-vaccinated animals. Non-vaccinated cattle undergo clinical and systemic disease in the early (acute) period lasting approximately 1–9 dpi while vaccinated cattle remain subclinically infected yet shedding virus between 1–7 dpi [85]. The transitional phase is associated with a reduction of clinical signs (if present) and either clearing of infection or ‘transitioning’ from early to persistent infection. The transitional phase in vaccinated and non-vaccinated animals occurs approximately between 7–14 dpi and 10–21 dpi, respectively. Entry into the persistent phase of infection is associated with subclinical FMDV replication in the nasopharygeal mucosa if the infection was not cleared in the transitional phase. FMDV genome copy number (GCN) estimates were calculated via qRT-PCR assay [24] and are provided for each sample with concurrent serum GCN estimates (Appendix A).

Antemortem samples collected from these animals included oral swabs, nasal swabs, serum, and oropharyngeal fluid (OPF) harvested using a probang cup [84]. Postmortem vesicular lesions (Ves) or nasopharygeal mucosa (Np) were collected at necropsy. Host factors including immunoglobulin and transcriptomic data were analyzed in prior publications [23,24,43]. The inoculum and a total of 103 samples from 20 animals were included in the present analysis; 77 samples originated from 10 non-vaccinated animals and 26 from 10 vaccinated animals.

### 4.2. Inoculum 

The FMDV A24 Cruzeiro (GenBank # SRP149342) inoculum was derived from a field strain, passaged once in BHK-21 cells and twice in cattle as previously described [16]. The virulence and viral dynamics of this inoculum were highly similar to virulent field strains of FMDV. The first bovine passage consisted of harvested vesicular epithelium and vesicular fluid obtained at 48 hours post tongue inoculation of two animals. The filtered suspension generated from the harvested material was subsequently used to inoculate a second cohort of three cattle. Vesicular fluid and epithelium were again harvested at 48 hours post inoculation and processed (macerated and filtered) to generate the virus suspension that was used to infect all animals in the present work. The inoculum was aliquoted and stored at −70 °C until use, at which time 2 mL of 10^5^ BTID_50_ (50% infectious does titrated in bovine tongue epithelium) [26] was used for inoculation in the current study. Previous deep sequence analysis of the inoculum indicated a complex population structure including 7 distinct viral clades [16]. In the current paper the inoculum was further resolved to identify 217 SNVs > 0.5% frequency (Appendix A). It was assumed that these frequencies were similar in all doses of the inoculum delivered to all cattle and that the full breadth of variants were available to seed infection.

### 4.3. Sample Passage and Sequencing

Illumina-derived deep sequence was examined for 75 of the total 103 virus samples (NCBI PRJNA473786). The consensus sequences of 52 of these samples were previously published, (GenBank MH426523-74) [16]. Nine samples were not passaged, noted as ‘raw’ in Appendix A and Appendix A, and 94 samples were passaged once in LFBK-αvβ6 cells [86]. A subset of samples was sequenced both with and without passage; SNV parity between these sample pairs supported the use of singular passage in LFBK cells (Appendix A). Viral RNA was extracted using the MagMAX RNA Isolation Kit (Thermo Fisher Scientific), reverse-transcribed and amplified, generating three overlapping amplicons covering the full CDS. Sequencing libraries were prepared with the Nextera XT DNA Library Prep Kit (Illumina, USA) and sequenced on the Illumina NextSeq 500 platform. All reads were quality-filtered, primer-trimmed and mapped to the inoculum consensus sequence in CLC Genomics Workbench v. 10 (www.qiagenbioinformatics.com). Read coverage of samples for which deep sequence was included in this study ranged from 338–155,000 (mean 30,000) averaged across the CDS. The inoculum deep sequence run totaled 10.4 million reads with a minimum coverage of 72,700 across the CDS.

### 4.4. Consensus-Level Sequence Analysis

Alignments, pairwise distances and the maximum likelihood phylogeny (PhyML 3.2 [30]) were evaluated in MEGA 7.0 [87] and Geneious 7.1 (www.geneious.com [88]). Substitution rates were calculated by tabulating pairwise nucleotide differences between each consensus sequence and the preceding sample sequence as a function of elapsed time between the two sample acquisitions. In cases for which there were multiple samples from the same animal on the same date (differing only by sample type or passage history), values were averaged. Statistical significance of differences between rates was calculated by way of two-tailed T-test.

### 4.5. Subconsensus Sequence Analysis

The Low Frequency Variant Detection tool in CLC Genomics Workbench was utilized to determine variant sites present within each deep-sequenced sample present in >2% of mapped reads with a minimum coverage of 20 reads and 0.75 strand-bias filter. Only two variants of the >1000 identified across all samples were called from coverage <100; and only four samples included regions with any coverage below 1000 across the CDS analyzed (Appendix A). While this variant detection pipeline was not specifically optimized for FMDV sample deep sequence variant calling, it is ploidy-independent and includes a sequencing error model test for significance. For the inoculum, variants present ≥0.5% were determined. Consensus-level sample substitutions that matched SNVs present at ≥0.5% in the inoculum deep sequence were categorized as ancestral SNPs, having most likely been present in an ancestral genome (i.e., transmitted in the inoculum gene pool). The remaining substitutions, those not detected in the inoculum ≥0.5%, were classified as novel SNPs, more likely to have resulted from within-host *de novo* mutation. Shannon entropy was calculated from quality-filtered and primer-trimmed reads in natural log units with a custom script.

### 4.6. Test of Diversifying Selection

Sites with evidence of having evolved under positive selective pressure in FMDV populations within hosts over time were determined with a mixed effects model of evolution (MEME) analysis in the HyPhy package [89]. MEME analysis of FMDV populations from each individual host identified sites encoding amino acid changes that significantly deviate from those that occur under neutral models of evolution; positively selected sites of statistical significance (*P* ≤ 0.10) are included in Figure 4. In order to incorporate subconsensus variation and linkage between low-frequency variants in MEME, haplotypes were reconstructed with the ViQuaS pipeline [39] with *SSAKE* [90] parameters o = 5, r = 0.75. This pipeline reconstructs the haplotypic composition present within each sample from quality-filtered, primer-trimmed Illumina reads. Because the depth of haplotype reconstructions of ViQuaS is greater than the practical capacity of MEME analysis, the frequencies of haplotypes utilized for downstream analysis were cut-off at either >0.5% for cattle 14-33, 14-49, 14-108, 14-111, 15-12, 15-13, and 15-14 or >2% for cattle 14-34 and 14-110. It is noteworthy that haplotype reconstruction may be susceptible to assembling both false recombinants and missing true recombinants. On this basis, in the present study, these haplotypes were used only for site-specific (MEME) analysis.

### 4.7. Haplotypic Composition of Sample Populations

In order to characterize FMDV lineages that made up each sample population, haplogroup-specific SNPs were first inferred from consensus sequence-derived phylogenetic relationships. The presence of these characteristic (haplotypic) SNPs dictated the subpopulation in which the sample was classified (Figure 3). Idealized criteria for these SNPs were: (i) shared with all members of a lineage, (ii) present in identical consensus sequences derived from different animals, (iii) detected at proportionate frequencies at the subconsensus level, and (iv) present in at least one homogeneous sample. For most samples, the dominant (majority) virus was identified by the sample consensus and location in the phylogeny. This approach is exemplified with sample 14-34, 6 dpi (Appendix A.), where haplogroup F characteristic SNPs (orange) are represented by approximately 79% of reads, group A (blue) includes SNPs ranging from 14.5%–20.3% and group C (green) by ~4.5% of reads. Relative haplotype frequencies within each sample were established using the lowest SNP frequency among each haplotype’s characteristic SNPs. These frequencies were used to construct proportional stacked bar graphs (Figure 3); samples lacking deep sequence data had 49% of the population designated undetermined. 

### 4.8. Protein Structure

The inoculum capsid protomer homology modeling was executed in SWISS-MODEL (swissmodel.expasy.org) with an FMDV A22 (PDB 4GH4) template. UCSF Chimera 1.13 (www.cgl.ucsf.edu/chimera [91]) was used for annotation and imaging. Annotation of antigenic regions and heparan sulfate binding site are based upon published works [32,35,36,37].

### 4.9. Data Availability

All new sequence data have been made available as sequence read archive (SRA) files at the National Center for Biotechnology Information (NCBI) under SAMN10280742-861. Previously published sequence data included in the present work is also available at NCBI, GenBank MH426523-74.

## Figures and Tables

**Figure 1 pathogens-09-00208-f001:**
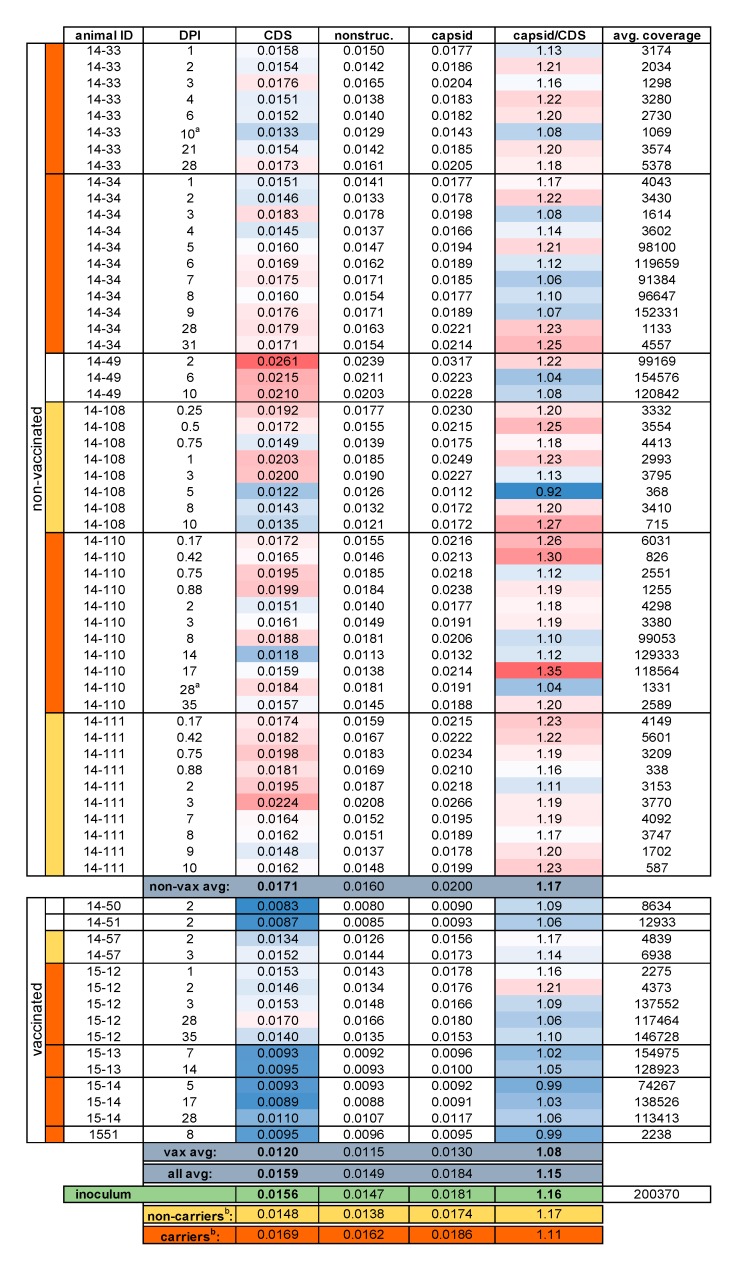
Average Shannon entropy in sample FMDV coding regions. DPI: Days post infection, CDS: Coding sequence, nonstruc.: Nonstructural protein coding regions (including VP4), capsid: VP1, VP2, and VP3 coding regions. Heat coloring indicates relative values with red denoting higher and blue denoting lower. ^a^ Non-vaccinated only, dpi-matched (5–14 dpi) samples averaged for non-carriers and carriers. Animals with undetermined carrier status (14–49, 14–50 and 14–51) were euthanized at 10 dpi or earlier. ^b^ Unpassaged samples.

**Figure 2 pathogens-09-00208-f002:**
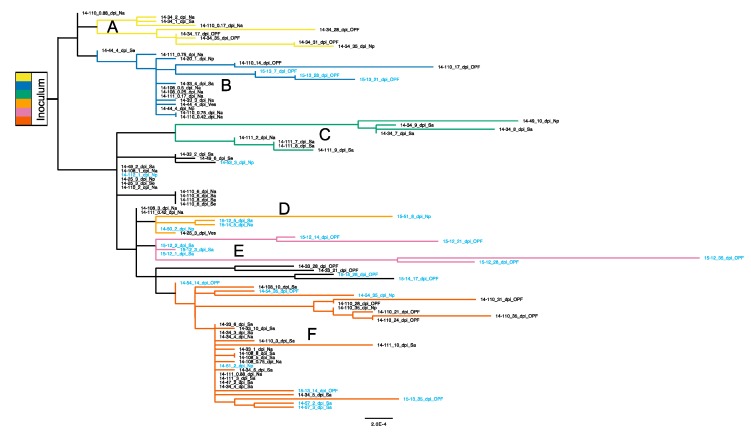
Inoculum-rooted maximum likelihood phylogeny of sample consensus sequences determined using PhyML [30]. Non-vaccinated cattle IDs are black text and vaccinated are light blue. Haplotypic clades (A–F) are colored according to inferred ancestral relationships and correlated polymorphism frequencies (see ‘haplotypic composition of sample populations’ in methods).

**Figure 3 pathogens-09-00208-f003:**
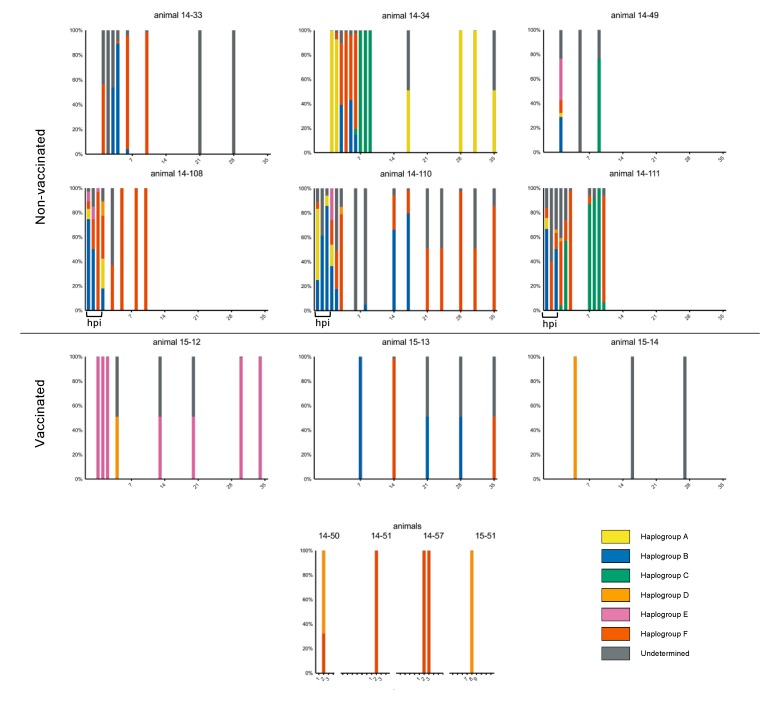
Haplogroup proportions in virus samples from nine serially-sampled cattle and colored as in Figure 2. Proportions were determined based on single nucleotide polymorphism (SNP) profiles characteristic of each lineage. Grey indicates lineage undetermined. X axis is days post infection at which sample was taken; ‘hpi’ is hours post infection, between 4 and 21 hpi. For persistent phase samples denoted as ‘undetermined’, this category has either been assigned because only consensus-level data was available (see Appendix A), in which case only >50% of variation was assignable to a specific haplogroup, or when the subconsensus variation was of unknown provenance.

**Figure 4 pathogens-09-00208-f004:**
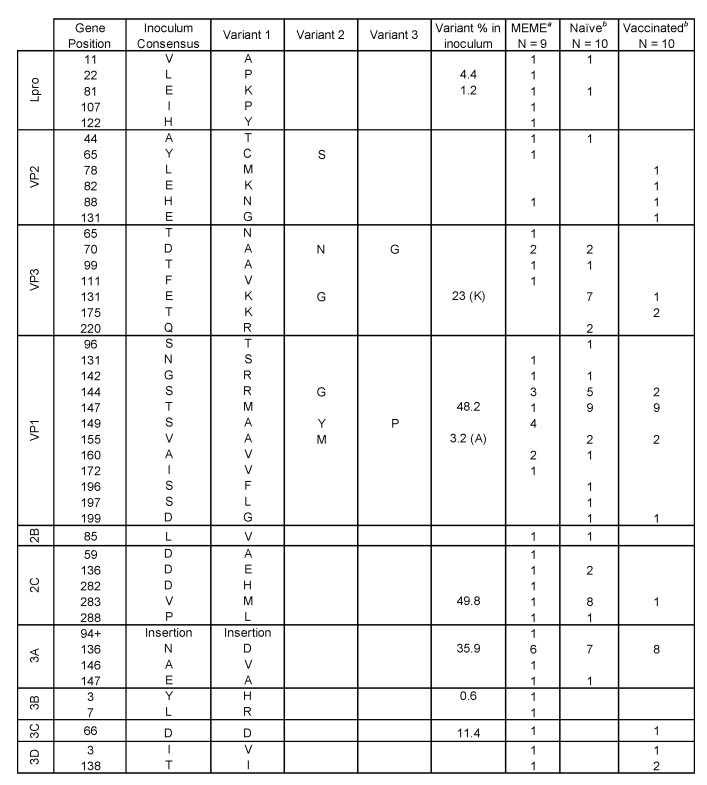
Variable FMDV capsid amino acids and positively-selected sites combined chart. *^a^* Presence of positively-selected sites identified in individual serially-sampled cattle using mixed effects model of evolution (MEME) analysis (*P* ≤ 0.10) (Murrell et al. 2012). *^b^* Amino acids identified as variable between viruses at the consensus level in VP1, VP2, and VP3 capsid coding regions.

**Figure 5 pathogens-09-00208-f005:**
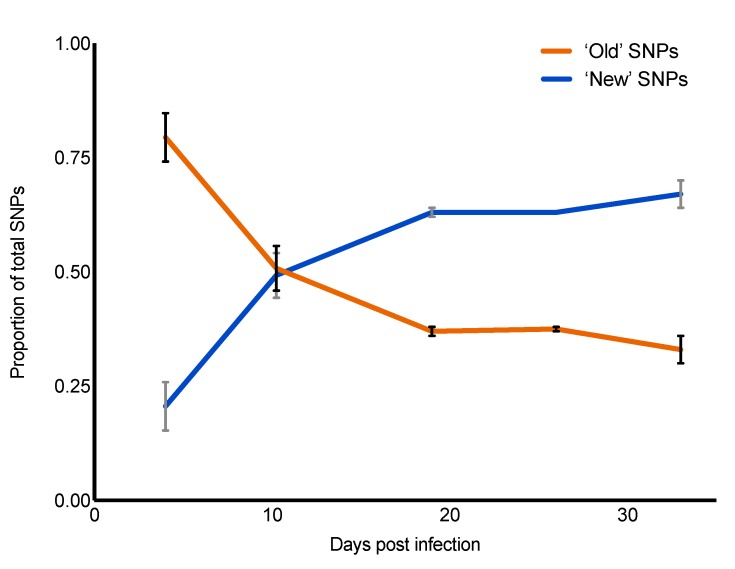
Proportion of inherited polymorphism. Substitutions present at the consensus level across samples also present at the subconsensus level in the inoculum (≥ 0.5%) are defined as ancestral SNPs. The remaining nucleotide changes observed in sample consensuses (not detected in the inoculum) are defined as novel.

**Figure 6 pathogens-09-00208-f006:**
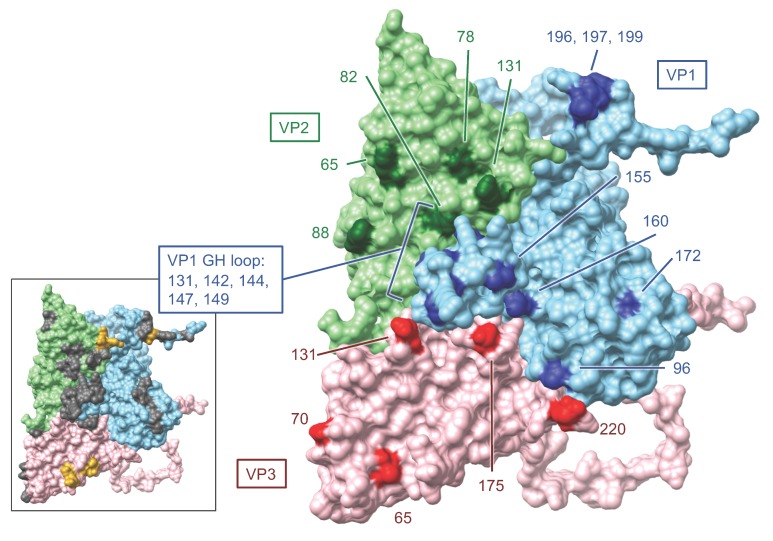
Homology model of inoculum consensus sequence with template PDB 4GH4 (FMDV A22). Surface view of FMDV A24 capsid protomer including VP1 (blue), VP2 (green), and VP3 (red). Labeled amino acid sites are those found to be under selective pressure according to MEME analysis as well as amino acids identified to be variable at the consensus level in samples derived from infected cattle. Inset: known antigenic sites in grey and heparan sulfate binding sites in yellow.

**Table 1 pathogens-09-00208-t001:** Nucleotide substitution rates and Shannon entropy for foot-and-mouth disease virus (FMDV) samples.

	Substitution Rate (subs/st/yr)			Shannon Entropy			
	Early	Transitional	Persistent	CDS	Nonstructural	Capsid	Capsid/CDS
inoculum	-	-	-	0.0156	0.0147	0.0181	1.16
non-vaccin. cattle	0.188*^a^*	0.127	0.080 *^a^*	0.0171*^b^*	0.0160 *^c^*	0.0200 *^d^*	1.17 *^e^*
vaccinated cattle	0.131	0.089	0.079	0.0120 *^b^*	0.0115 *^c^*	0.0130 *^d^*	1.08 *^e^*

*^a^* = *P* < 0.05; *^b, c, d, e^* = *P* < 0.001; subs/st/yr: Substitutions/site/year. CDS: Coding region.

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
