# Peer review of "Into the Deep (Sequence) of the Foot-and-Mouth Disease Virus Gene Pool: Bottlenecks and Adaptation during Infection in Naïve and Vaccinated Cattle"

_pathogens, 2020, doi:10.3390/pathogens9030208_

Round 1

Reviewer 1 Report

In the submitted manuscript Fish et al. described consensus and sub consensus sequence analysis of FMDV samples collected during a longitudinal  FMDV experimental infection of vaccinated and unvaccinated cattle, which is a follow up from previously published work by the same group. In this study virial variants found in the inoculum (which is characterised by presence of multiple haplogroups) dominate FMDV sequences found in the collected samples (both from unvaccinated and vaccinated animals) especially at the early stage of infection, with shift in haplogroup dominance present over time of infection. While there was no vaccination-specific substitutions found, there variance and entropy are lower in the vaccinated cohort, suggesting strong bottleneck granted by the vaccination. In both animal cohort non-synonymous substitutions were found, and especially those found in the capsid appeared to be driven by selection process e.g. SGD to RGD. Number of escape mutants were found in vaccinated animals. While entropy levels were maintained to the persistent phase of infection, in most cases only one haplotype dominated with numerous minority variants. There was no significant difference in substitution analysis between carriers and terminators. 

Overall the article is interesting and brings new findings which are of interest to scientific community.

I am a bit sceptical that one will obtain exact single nucleotide variants (SNVs) from viral samples obtained directly from an animal and from a single cell passage. I assume that when performing the passage authors waited for occurrence of CPE which means several rounds of viral replication, and there is an evidence that virial sequence space expands when RNA viruses encounter new environment. While used here LFBK express integrin, they are of pig origin which I consider a substantial change of environment for the virus. Authors mentioned ‘SNV parity’ between ‘raw’ and passaged samples. Can authors please present relevant data showing correlation and its extend of SNVs from relevant passaged and not-passaged samples.

To my understanding Low Frequency Variant Detection tool implemented in CLC Genomic Workbench was initially designed for mammalian samples such as cancer cells or mixed tissues. This means that the statistics implemented in this tool might not be adequate for variant calling in RNA virial samples since the later have much higher allele frequencies and hence error detection will be more difficult. Personally I would re-analyse data in a variant caller which was designed specifically for virial samples. I know that for many samples authors looked only at variant above 2%, however there is evidence that some errors can be found even at that level of frequencies. Just something worth considering or at least discussing. I also would not recommend carrying out any variant analysis on sites showing read depth lower than few hundreds (authors used sites with read depths of 20).  

Figure 2: Very similar data from this animal study were presented in previously submitted by this group manuscript (Arzt et al. 2019). I don’t think Figure 2 adds novel finding to this manuscript and could perhaps be moved to the supplementary data.

Figure 3 and relevant data: I am sceptical about how much conclusion can be made about the haplogroup diversity (or its lack) in the persistent stage of infection from data shown in figure 3. From a total of six unvaccinated animals only three animals show data for persistent infection time points, where one animal shows only ’undetermined’ haplogroup (it is unclear to a reader whether that means a single or multiple haplogroups) and two others show mixture of singular ancestral haplogroup and ‘undetermined’ haplogroup(s). Similar situation is for vaccinated animals. 

Figure 6 and relevant data: While I appreciate that analysis of the sites under selective pressure is novel for the dataset, it should be mentioned that some of the described here substitutions were already described in Arzt et al. 2019. 

I had no access to supplementary data since link provided in the manuscript did not work and not all of the supplementary data appeared to be present at the manuscript submitted to https://www.biorxiv.org.

It seems that in their haplotype reconstruction authors do not seem to consider events of recombination which appears to be very common for FMDV. While I appreciate that in this study recombination would be very difficult/ impossible to detect, and that some approximation needs to be adapted when performing haplotype reconstruction, one should also discuss limitation in such analysis especially that it is possible that two different SNPs appearing at the same frequencies but not existing at the same read can originates from different genomes.

Lines 502-4: I wonder whether that conclusion might be a bit over simplistic. One needs to remember that in here we only study genomes of the virus and not all synthesised genomes are packaged (actually there is some evidence suggesting that only small proportion of the synthesised genomes are packaged) and genomes are not packaged in capsids they encoded. That scenarios could also be considered. 

Reviewer 2 Report

The authors infected  vaccinated and non vaccinated cattle with (A24-Cruzeiro) and 35 days post infection they analyzed the infection by deep-sequencing to identify viral population during the corse of infection.

Questions:

1- why do you think that the non vaccinated cattle had minimal immunological response against the virus? How did you measure their immunological response after the infection?

2- What was the virulence of the virus you used in your experiment? and how different do you think it is from the virus in the naturally occurred cases?

3- Current FMD vaccine is effective in protecting the cattle and preventing viral adaptations. In what way your data can help in improving the vaccine ?

4- The discussion section is very long and require some editing.

Reviewer 3 Report

The manuscript builds upon previous work that found differences in virus substitution rates between vaccinated and unvaccinated cattle across the progression phases of the disease to show that these differences are attributed to narrower bottlenecks of FMDV early on and novel mutations associated with subclinical infection.  The manuscript provides interesting contributions to our understanding of the FMD virus and offers potentially important implications for control with vaccination. The structure and content is well-presented. I only have three comments and two minor suggestions for revision:

Comments:

Results

  1. Line 111-By mentioning that your method of measuring genomic change is inconsistent with the dynamics of FMDV and there are no commonly accepted methods, I then become curious as to what other methods are used or how this question is approached in other studies. Further clarification as to how this relates to previous publications is needed.

Discussion

  1. This manuscript builds on previous work (17). In the second paragraph, lines 360-370, the discussion on bottlenecks seems the most appropriate place to re-emphasize how this manuscript builds on (17), specifically with respect to during persistent infection.

Conclusion

  1. Line 514-516—The conclusion states that “differential quasispecies behavior in vaccinated hosts may provide insights into further enhancement of countermeasures to impede viral propagation.” Additional specification of these enhancements of countermeasures would be beneficial. You introduce the manuscript by indicating that vaccination campaigns to control FMD can be a burden. This would be an ideal time to place this research into the broader context of improving FMD vaccination efforts, ie as to whether your findings provide an indication of the timing of vaccination, quarantine, or the duration of spread.

Minor Suggestions:

  1. Line 272—Specify whether these cutoff points for the relative frequencies of the haplotypes are standard or represent your sensitivity analyses around potential cutoff ranges.
  2. Line 258-259—Figure 5; making one of the lines dashed or changing one of the colors would better distinguish between the two.

Reviewer 4 Report

The study describes a deep sequencing analysis of FMDV samples experimentally administered to vaccinated and non-vaccinated cattle. Differences, similarities among populations and sub-consensus evolution of several genomic regions are examined. The data obtained is relevant to the understanding of FMDV evolutionary mechanisms although the study and its conclusions have several limitations that are not properly addressed.

  1. The main problem with this study is the complexity of the inoculum. No matter how detailed was the description in previous publications, in the present one the authors should give additional information on the volume inoculated into each animal and the types of variants that (given their frequency) are likely to enter all animals, or some of the animals, or none of the animals. This data should be provided in Materials and Methods and the limitation represented by a complex inoculum should be included in the Discussion.
  2. Provide quantitative information of the estimate of viral loads in the infected animals.
  3. It is not clear if the evidence of bottlenecks is related to antibody selection in vaccinated animals or the authors speak of two different events (bottlenecks separate from selection).
  4. It is not clear either why fitness differences between variants should be minimized with large population sizes since completion among variants should be favored.
  5. The reference to Wright C F et al. J. Virol. 2011 on deep sequencing of FMDV in cattle should be quoted.
  6. There is considerable room to render a more concise text. Some introductory references can be replaced by the more recent book edited by Sobrino and Domingo (Caister Academic Press, Norfolk, UK, 2017) that includes articles by authors from several schools on most aspects of FMD and FMDV. References 6 and 18 are the same.

Round 2

Reviewer 1 Report

I would like to thank the authors to take their time and answer my concerns.

While I not necessarily fully agree with the variant detection approach, I am happy to accept the used methodology for the purpose of this manuscript and accept authors response. 

Figure 3 (lines 220-3)
Thank you very much for explaining he classification as undetermined. I do understand the limitation of the study and sampling. However I still think that since authors show that only three animals out of nine  showed decreased within-host haplotypic diversity (while five animals lack data for persistent phase of infection an animal 15-13 shows switching between haplogroup B and F) it is an overstatement to say “As both animal cohorts (vaccinated and non-vaccinated) progressed to the persistent phase of infection, within-host haplotypic diversity decreased, with each persistent phase sample containing viruses belonging to only a single haplogroup (Figure 3)”. Authors simply don’t know what happens in 4 animals for which authors do not have data for (or at least these data are not presented).  I am happy with the authors changing the cited statement to something like that “considering presented data it is possible that as both animal cohorts (vaccinated and nonvaccinated) progressed to the persistent phase of infection, within-host haplotypic diversity decreased….”. 

Line 227 and 230 authors give Table S1 as table which present data for subconsensus polymorphism. I saw only two supplementary materials Figure S1 which contains a table which does not show subconsensus polymorphism data and Table S3 which presents data for comparison of SNPs between passaged and un-passaged viruses. Is there some supplementary data missing? 

Thank you for completing all the other corrections. Once the two items are corrected above, I am happy with the manuscript to be accepted. 

Reviewer 4 Report

the authors have addressed satisfactorily my concerns
